# DNA Methylation in Neurodegenerative and Cerebrovascular Disorders

**DOI:** 10.3390/ijms21062220

**Published:** 2020-03-23

**Authors:** Olaia Martínez-Iglesias, Iván Carrera, Juan Carlos Carril, Lucía Fernández-Novoa, Natalia Cacabelos, Ramón Cacabelos

**Affiliations:** Euroespes Biomedical Research Center, International Center of Neuroscience and Genomic Medicine, 15165 Corunna, Spain; biotecnologiasalud@ebiotec.com (I.C.); genomica@euroespes.com (J.C.C.); genetica@ebiotec.com (L.F.-N.); serviciodocumentacion@euroespes.com (N.C.); rcacabelos@euroespes.com (R.C.)

**Keywords:** Alzheimer disease, Parkinson disease, DNA methylation, neurodegenerative diseases, cerebrovascular diseases, DNMTs, diagnostic biomarker

## Abstract

DNA methylation is an epigenetic mechanism by which methyl groups are added to DNA, playing a crucial role in gene expression regulation. The aim of the present study is to compare methylation status of healthy subjects with that of patients with Alzheimer’s, Parkinson’s or Cerebrovascular diseases. We also analyze methylation status of a transgenic Alzheimer’s disease mouse model (3xTg-AD). Our results show that both global methylation (*n* = 141) and hydroxymethylation (*n* = 131) levels are reduced in DNA samples from buffy coats of patients with neurodegenerative disorders and age-related cerebrovascular disease. The importance of methylation and hydroxymethylation reduction is stressed by the finding that DNMT3a mRNA levels are also downregulated in buffy coats of patients with Dementia (*n* = 25). Global methylation is also reduced in brain, liver and serum samples of 3xTg-AD vs. wild type mice, such as DNMT3a mRNA levels that are also decreased in the brain of 3xTg-AD (*n* = 10). These results suggest that the use of global methylation and hydroxymethylation levels, together with the study of DNMT3a expression, could be useful as a new diagnostic biomarker for these prevalent disorders.

## 1. Introduction

Neurodegenerative and cerebrovascular diseases represent major health problems in Western countries and are usually linked to aging. Both are complex and multifactorial disorders and are caused by a combination of genetic and environmental factors. The most relevant neurodegenerative disorders worldwide are Alzheimer´s disease (AD) and Parkinson´s disease (PD). Cerebrovascular diseases can also result in Vascular Dementia as a consequence of vascular-related brain tissue damage. About 40–50 million people suffer dementia and it is thought that this number will increase to 145 million in 2050 [1].

Epigenetics study heritable alterations in gene expression that happens without any change of the DNA sequence, linking the genome with the surrounding environment [2,3,4]. It has been well established that the accumulation of different epigenetic alterations over the lifespan might contribute to neurodegenerative and cerebrovascular disorders and that neuroepigenetics plays a crucial role in neurodegenerative processes [5,6,7]. DNA methylation is the most investigated epigenetic hallmark. It is a reversible mechanism in which methyl groups are added to cytosines located in CpG (5′-Cytosine-phosphate-guanosine-3′) nucleotides turning these cytosines into five methylcytosines (5mC). This process changes DNA stability and accessibility, regulating gene expression [8]. Hydroxymetylation (5hmC) is another epigenetic mechanism that modifies five methylcytosines, adding a hydroxymethyl group. This epigenetic mark is especially frequent in brain cells and it has been suggested that it is both an intermediate state in the demethylation process and a key epigenetic mark in neurological disorders [9,10,11]. Hydroxymethylation is implicated in important processes such as differentiation and gene expression regulation.

Usually, DNA methylation is a repressive mark [12], which attracts other silencing elements such as methyl-CpG-binding proteins [13,14]. The addition of methyl groups is carried out by DNA methyltransferases (DNMTs) [15]. There are three DNMTs family proteins: DNMT1, DNMT2 and DNMT3. All of them are expressed in neurons [16] with different functions. DNMT1 is responsible for maintaining the methylation pattern after cell division, being responsible for the inheritance of methylation marks [17]. DNMT3a and DNMT3b are responsible for de novo methylation [18,19]. Furthermore, Ten-Eleven Translocation (TET) proteins oxidate the methyl group and converts 5mC into 5-hydroxymethylcytosines (5hmC) [20].

The search of reliable biomarkers for neurodegenerative and cerebrovascular diseases might help to an early diagnosis and the implementation of a precision medicine program with personalized treatments. At this moment, there are not appropriate and reliable epigenetic biomarkers for diagnosis, classification and disease progression [21]. Indeed, most of the current biomarkers are based on expensive and/or invasive techniques such as neuroimaging methods or cerebrospinal fluid (CSF) analysis [22]. Liquid biopsy is a cheaper and more comfortable tool. The search of epigenetic biomarkers in accessible fluids such as blood has focused several research lines in the past years; however, no reliable epigenetic biomarkers are still available for neurodegenerative and cerebrovascular diseases. 5mC has been proposed as a new biomarker with contradictory results. Further studies are needed to clarify the possible use of 5mC and 5hmC as biomarkers of different neurodegenerative and cerebrovascular disorders. In this study, we analyze global 5mC, global 5hmC and DNMTs expression in Parkinson’s (PD), Alzheimer’s (AD) and Vascular Dementia (DV), comparing values with those obtained from healthy individuals. This study is completed with the analysis of 5mC levels and DNMT3a expression in an AD mouse model. Our results provide valuable information about the DNA methylation status in these prevalent pathologies. The complete study of methylation may provide a useful tool for the diagnosis and monitoring of patients with age-related neurodegenerative and cerebrovascular disorders.

## 2. Results

### 2.1. Global DNA Methylation Decreases in Peripherally Blood from AD, PD and DV Patients

In order to analyze possible changes in DNA methylation between healthy subjects and patients with neurodegenerative or cerebrovascular disorders, global 5mC levels were measured in buffy coat samples. For this study, we used DNA from 40 healthy subjects and 40 patients with AD, 35 with PD and 26 with DV. The characteristics of the patients included in this study are shown in Table 1. Samples from healthy individuals showed a medium value of 5mC of 4.14% ± 0.38. In samples from AD and PD patients, these values fell to 2.51% ± 0.2 and 2.41% ± 0.24, respectively. Finally, in DV patients, medium values obtained of 5mC were 2.54% ± 0.29. In summary, global methylation levels were found to be significantly lower in the three pathologies in comparison to values obtained in DNA samples from healthy subjects (Figure 1). The histograms of the data obtained in AD, PD and DV groups are shown in Appendix A and normality test results in Appendix A.

Given the decrease in 5mC levels observed in the pathological samples, we carried out linear regression studies to analyze whether there were any correlation between 5mC levels and different parameters such as age, sex or even psychometric assessment (Appendix A). We only find a significant positive correlation between age and 5mC levels in PD (Appendix A) with a *p*-value of 0.0385. We did not observe any correlation between global methylation values and any of these parameters, neither in healthy subjects nor in AD or DV groups. The *p*-value of the correlation between age and 5mC levels in DV group was 0.0752. Since the APOE genotype is the main genetic risk factor for AD, we also analyzed the possible link between global methylation levels and the different variants of the *Apolipoprotein E* (*APOE*) gene. We did not find any correlation between these parameters (Appendix A).

### 2.2. Patients with AD, PD and DV Show Lower Global DNA Hydroxymethylation Levels in Peripherally Blood

There is less bibliography about the role of 5hmC as an epigenetic mark. It is not only an intermediate product produced in the removal process of 5mC but also an epigenetic modification by itself, regulating gene expression. 5hmC is involved in neurodevelopment and also in neurodegenerative diseases. For these reasons, we measured DNA 5hmC levels in buffy coat samples from 40 healthy subjects and 47 patients with AD, 30 with PD and 21 with DV. Characteristics of patients are shown in Table 1. 5hmC values were found to be significantly lower in the three pathologies compared to healthy subjects. However, the reduction was higher in DNA from Parkinson’s disease patients, with a medium value of 0.08% ± 0.01 versus the 0.2% ± 0.02 obtained in healthy subjects. In DNA samples from Alzheimer’s disease or Vascular Dementia, 5hmC values were 0.15% ± 0.02 and 0.13 ± 0.02, respectively (Figure 2). The histograms of the data obtained in AD, PD and DV groups are shown in Appendix A and normality test results in Appendix A.

We also analyzed if there was a correlation between 5hmC levels and different parameters such as age, sex or psychometric parameters and did not observe any correlation between them in healthy subjects nor in any of the disease groups (Appendix A). However, *p*-values obtained in the correlation between age and 5hmC levels in PD and DV were close to significant (*p*-values of 0.07857 and 0.0752, respectively). We neither observed any correlation between APOE variants and 5hmC levels (Appendix A).

As we observed regulation in both 5mC and 5hmC levels in the pathologies studied, we decided to perform correlation studies between both values. We only found a significant correlation between 5mC and 5hmC values in AD group, with a *p*-value of 0.0425 (Figure 3). Interestingly, the correlation between 5mC and 5mhC was negative in healthy samples and positive in all the pathological groups (although only statistically significant in AD).

### 2.3. DNMT3a Expression Decreases in Blood Samples of Patients with Dementia

DNMTs are the main enzymes involved in the DNA methylation process. Given the decrease observed in global DNA methylation in AD, PD and DV patients, we also analyzed DNMT expression by real time retrotranscription-polymerase chain reaction (RT-qPCR) in blood samples of seven healthy subjects. We compared these levels with those obtained in 10 samples from PD patients and eight samples from patients with different types of Dementia. Characteristics of samples are shown in Table 1.

DNMT1 is responsible for maintaining methylation patterns after DNA replication, being responsible for the inheritance of methylation marks. We did not detect any significant change in DNMT1 expression (Figure 4A) between healthy subjects and PD or Dementia groups.

DNMT3a acts as a de novo methylation enzyme, rather than maintaining methylation of DNMT1. We analyzed DNMT3a expression in blood samples from the three groups. We did not find any difference between DNMT3a expression in healthy subjects and PD patients. However, DNMT3a expression was significantly downregulated in patients with dementia. This expression was reduced over 60% in blood samples from patients with Dementia (Figure 4B). Normality test results are shown in Appendix A.

We also analyzed if there was a correlation between DNMT3a expression and different parameters such as age, sex or psychometric parameters (Appendix A). DNMT3a expression was significantly higher in females than in males in PD samples with a *p*-value of 0.0087. In samples from patients with dementia, DNMT3a expression was lower in females, but without significant differences (*p*-value 0.0775). We observed an increase in DNMT3a expression in APOE 3.4 genotype vs. APOE 3.3. (*p*-value 0.039). We cannot analyze the signification of differences with the APOE 2.4 genotype because we only had one sample with this genotype. We did not observe any correlation between DNMT3a expression and the other parameters analyzed (Appendix A).

In our study, patients with dementia included Senile Dementia, Mixed Dementia, Alzheimer disease and Vascular Dementia. Detailed characteristics of each type of dementia are specified in Table 1. Given the heterogeneity of this group of patients, we analyzed the differences in DNMT3a expression between different types of Dementia and did not observe significant differences in DNMT3a expression between Senile Dementia, Mixed Dementia, Alzheimer disease or Vascular Dementia.

### 2.4. Global DNA Methylation Decreases in a Transgenic Mouse Model of Alzheimer’s Disease

In the light of the finding that 5mC levels were reduced in human DNA from AD patients, we wondered if global methylation was also regulated in a transgenic mice AD model. We used a triple transgenic mice model (APP/BIN1/COPS5), which shows Aβ deposits and severe deficits in synaptic plasticity [23]. We observed a significantly decrease in 5mC levels, either in the brain and liver of 3xTg-AD mice (Figure 5A) in comparison to values obtained in DNA samples from wild type mice. In brain samples, the medium value of global DNA methylation in wild type mice was 3.26% ± 0.12 and this value decreased to 2.65% ± 0.2 in 3xTg-AD mice. In liver samples, 5mC values were 2.27% ± 0.17 in wild type mice and 1.55 ± 0.14 in transgenic mice. We also analyzed 5mC levels in serum samples from these mice. Global methylation levels were also significantly reduced from 3.05% ± 0.1 in wild type to 2.14% ± 0.44 in 3xTg-AD mice (Figure 5B). As we observed a decrease in DNMT3a expression in human blood samples of patients with Dementia, we also analyzed the expression of this gene in 3xTg-AD mice brain. Consistent with our previous results, we observed a significant reduction of over 40% in DNMT3a expression in the brain of 3xTg-AD mice (Figure 5C).

## 3. Discussion

Genetic factors do not describe completely the origin and evolution of neurodegenerative and cerebrovascular diseases. Indeed, there is increasing evidence for the importance of environmental factors in the development and progression of these diseases. On the other hand, environmental effects on gene expression are mediated at least in part by different epigenetic mechanisms. DNA methylation is one of the most important epigenetic mechanisms. In this study, we analyzed global methylation and hydroxymethylation levels in neurodegenerative and cerebrovascular diseases. Blood DNA analysis is a non-invasive and cheap liquid biopsy technique, with potential value for clinical diagnosis. Finding new non-invasive biomarkers for diagnosing these diseases would be very useful in the management of these patients. Interestingly, Masliah et al. published that methylation levels are concordant in brain and blood samples from patients with PD [24]. In our study, we have analyzed 5mC and 5hmC levels in buffy coat samples from healthy subjects, comparing these results with those obtained from patients with PD, AD and DV. DNA methylation is currently a reliable biomarker in different diseases [25]; however, unfortunately, there are inconclusive results for neurodegenerative or cerebrovascular disorders. Our results show that global 5mC and 5hmC levels are significantly lower in brain disorders than in healthy samples. In the case of 5hmC quantification, the decrease is even more marked in samples from PD patients. Results of 5hmC levels in AD patients were quite scattered, probably by the heterogeneity of this disease. DV group includes mixed dementia, ischemic vascular encephalopathies, multi-infarction dementias, Bingswanger disease and other similar pathologies. This heterogeneity might produce a high variability in the results. Our results agree with different studies in which different authors show that global DNA methylation changes with aging and that these changes are linked to certain neurodegenerative diseases [24,26,27,28]. Different authors have described a decrease of 5mC or 5hmC levels in brains and blood samples in different animal models and human samples of these disorders [28,29,30,31,32,33,34,35]. Indeed, Aβ1-42 peptide reduces 5hmC levels in vitro [36]. Different brain regions have different 5hmC levels, with reduced 5hmC levels in the hippocampus but not in the cortex or cerebellum [36]. High hydroxymethylation regions are enriched in introns, exons and intergenic regions, and genes associated with these hydroxymethylated regions are implicated in neuronal development/differentiation and neuronal function/survival pathways [36,37].

However, there are also studies showing opposite results, without any significant differences between healthy and AD brain samples [35,38,39,40,41,42,43], or even observing an increase of global methylation and hydroxymethylation levels in different regions of the brain from subjects with AD [35,44,45,46,47,48,49]. These differences might be explained due to the study of different brain regions. Another explanation could be the heterogeneity of the pathological diagnosis of the analyzed samples. In the case of blood samples, some studies were performed in serum and other in leukocyte samples. This heterogeneity in the sampling might also explain the variability obtained in the results. In several cases, the number of samples studied was too low to obtain conclusive results. The differences in 5mC and 5hmC levels in different brain regions have also been well documented by Phipps et al., who showed low 5mC and 5hmC levels in astrocytes from AD patients vs. healthy samples. However, they have not found any difference in disease-resistant calretinin interneurons or microglia. These authors did not observe any differences near the plaques or in plaque-free regions in late-AD samples. However, they found higher 5mC and 5hmC levels in neurofibrillary tangles [49].

It has also been reported that global methylation changes with age [29,30,33,50]. In our case, we only found a significant correlation between age and DNA methylation in PD subjects. *p*-values obtained in the comparison of 5mC and age were close to significant in DV group. The correlation between 5hmC levels and age was close to significant in PD and DV subjects. 5mC may increase gradually after damage onset after 60 years of age [50], and most of our samples belong to patients older than 60 years. Thus, the age of patients used in our experiments could explain these differences. Possibly, the high number of samples and variety in the age of the subjects may have caused a higher significant correlation between 5mC and 5hmC and age. On the other hand, we did not find any correlation between sex, psychometric parameters or APOE genotype and global methylation or hydroxymethylation levels. In contrast, an increase in global methylation levels has been reported in AD patients harboring the APOE4 genotype [28]. Higher global methylation levels in whole blood from AD subjects and a correlation between global methylation levels and psychometric parameters was also reported by the same authors [28]. We did not find significant correlation between 5mC or 5hmC psychometric parameters or APOE genotype in AD samples. However, 5hmC levels were higher in APOE4 genotype, but without a significant difference due to the high standard error observed in APOE4 group of subjects. 5mC levels were also slightly increased in APOE4 subjects in the AD group. They analyzed global methylation by a luminescent assay, whereas we have used an ELISA-like colorimetric assay. They studied methylation in whole blood samples, which contain different cell types with different methylation profiles [51]. The analysis of whole blood instead of the buffy coat samples used in our study, together with the method of methylation quantification, might explain the differences between both studies.

There is limited information about the role of global methylation and hydroxymethylation in PD subjects. In a rat model, Zhang et al. studied 5hmC levels in the striatum and found an increase in 5hmC levels with aging, but they did not observe any difference in 6-OHDA treated versus no-treated rats [52,53]. Recently, a genome-wide methylation study in blood samples identified several genes in which methylation status changed during PD progression and also in response to treatment with dopaminergic medication [54].

There is little evidence about the relation between global methylation or hydroxymethylation in dementia associated with cerebrovascular diseases. For example, ischemia-hypoxia condition in astrocyte cultures increases global DNA methylation. Non-dividing cells, such as neurons, do not show any change in global methylation levels in ischemia-hypoxia conditions [55]. Chronic hypoxia in rat brains induces global DNA hypermethylation and also increases DNMT3a expression [56]. Increased levels of global methylation have been observed in men with high risk of ischemic stroke [57]. However, Soriano-Tárraga et al. did not find any significant changes in global DNA methylation in different stroke subtypes [58]. Conversely, and in accordance with our results, lower methylation levels were observed in stroke [57]. Another group obtained similar results in a cohort of patients in whom hypomethylation was associated with a high mortality risk [59]. Referring to hydroxymethylation, the results obtained by Tsenkina et al. indicated that 5hmC show a cell-type-specific pattern in a mouse model [60].

As DNMTs are responsible for DNA methylation, its activity and expression are very important for global methylation levels. We do not observe any difference in DNMT1 expression between healthy and pathological groups. Interestingly, a significant decrease in DNMT3a expression is observed in blood samples from patients with different types of dementia. However, we did not find any difference in DNMT3a expression between the different types of dementia. In PD subjects, DNMT3a expression was significantly higher in females than in males. Conversely, DNMT3a expression was lower in females in DV subjects. We should highlight the low number of samples used in the analysis of DNMT3a expression and a study with a greater number of samples may be necessary to obtain reliable conclusions. On the other hand, APOE3.4 genotype subjects showed higher 5mC levels than APOE3.3 genotype in DV group. We did not observe any correlation between age and psychometric analysis and DNMT3a expression. The first evidence on the potential relationship between DNA methylation and neurodegenerative diseases comes from the high DNMT levels detected in brain tissues. DNMTs are closely related to learning and memory functions [61,62,63,64]; and a decrease of DNMTs expression has been described during aging. Further evidence about the importance of DNMTs in the aging process is the fact that the hippocampus of aged mice shows a decrease in DNMT3a expression. DNMT3a overexpression reverses spatial memory deficits [65,66]. There is also evidence about the relation between DNMT expression and AD. For example, a decrease in DNMT expression in neurons and hippocampus of AD patients has been described [32,33]. Conversely, DNMT1 expression, at an mRNA and protein level, is increased in late-onset AD in blood mononuclear cells [28]. Once again, the differences between studies may be explained by the fact that different cells or regions are analyzed in different studies with diverse methodologies. Indeed, the study of Francesco et al. was performed only in patients with late-onset AD, and our study included different types of dementia in different stages of the disease. It has been also described that DNMT expression is also increased by folic acid, a crucial vitamin for the methylation process, in N2a-APP cells, and treatment with zebalurine, a DNMT inhibitor, increases Aβ production [67]. These results reinforce the important role that global methylation and DNMT expression may play in neurodegenerative diseases. Lower nuclear DNMT1 levels have been reported in postmortem PD brain samples. Similar results were obtained in an α-synuclein transgenic mouse model [68]. Translocation of DNMT1 from the nucleus to the cytoplasm may lead to global hypomethylation in human and mice brains [68]. In contrast, the group of Endres et al. showed that a reduction in DNMT1 expression in a transgenic mouse model, but not its absence, may lead to a protection of post-mitotic neurons from ischemic brain injury [69]. Indeed, the same authors showed that ischemia leads to an increase in DNA methylation and also that low DNMT levels provide a brain protective effect against ischemic injury [70]. Furthermore, Lee et al. suggested that DNMT1 down-regulation after an ischemic event may be related to neuronal death [71]. In rat brain models, hypoxia causes a decrease of global methylation and DNMT1 and DNMT3a expression [72].

The distinction between different neurodegenerative pathologies is difficult because similar processes are altered in different neurodegenerative diseases [73]. One example is the amyloid plaque, one of the main characteristic hallmarks of AD, which is also present in dementia with Lewy bodies (LBD) or early Alzheimer-like profile linked to Down’s syndrome (DS) [74]. Tau-related neurofibrillary tangles are also found in PD and DLB [75]. Finding a suitable biomarker that allows clinicians to distinguish between the different neurodegenerative disorders would be very useful in the management of these patients. In this study, we propose methylation and hydroxymetylation studies to increase reliability in the differential diagnosis and follow up of age-related brain disorders. As far as we know, there are few studies where methylation and hydroxymethylation levels are analyzed. Interestingly, we have observed a significant positive correlation between 5mC and 5hmC values in AD subjects. This analysis may not only be useful as a novel diagnosis biomarker of neurodegenerative and cerebrovascular diseases but it may also help to distinguish patients with PD (with lower 5hmC levels and no changes in DNMT3a expression) from AD or DV. Thus, whole study of 5mC, 5hmC and DNMT3a mRNA levels may give to the clinician a global vision of the methylation status of the patient, improving its ability for early diagnosis and future drug monitoring.

## 4. Materials and Methods

### 4.1. Subjects

Blood samples from AD, PD, DV and healthy cases were obtained from the CIBE collection (C000925, 21 October 2013, EuroEspes Biomedical Research Center). This collection follows ethical procedures according to the Spanish regulation (Ley Orgánica de Investigación Biomédica, 14 July 2007) and was obtained after informed consent from all patients and/or legal caregivers. Patients were selected to show similar distributions of age, sex and APOE genotype and its characteristics are showed in Table 1. Diagnosis of patients was done by neurologists, after a thorough clinical and genomic study, and classified according to internationally accepted diagnostic criteria. Patient’s study includes the study of several SNPs linked to PD, AD or vascular risk, psychology tests, brain mapping and neuroimaging proofs. The results of these proofs were analyzed by a neurologist.

From all cases we collected an EDTA-coated tube from peripheral blood and after centrifugation the buffy coat was stored at −40 °C until DNA extraction. In the case of RNA, samples were subjected to a first erythrocyte lysis step and were centrifuged to precipitate lymphocytes. Then, lymphocytes were homogenized into Qiazol Lysis Reagent (Qiagen, Hilden, Germany) and stored at −40 °C until its use.

### 4.2. Mouse Model

Triple-transgenic 3xTg-AD mice (APP/BIN1/COPS5) overexpress the Swedish mutation of APP (human amyloid precursor protein), BIN1 (bridging integrator 1, AMPH2) and COPS5 (COP9 constitutive photomorphogenic homolog subunit 5, Jab1). Mice were kindly donated by Dr Laksmana’s lab and backcross-bed in our laboratory. All experimental procedures were performed in compliance with the European Community Law (86/609/EEC), EU Directive 2016/63/EU and the Spanish law (R.D. 1201/2005), with approval of the Ethics Committee of the Euroespes Research Centre (Permit number: EE/2015-184).

Eight–Eleven-week-old wild type C57BL/6J or 3xTg-AD mice were sacrificed and liver and brain samples were collected. We also collected blood samples by cardiac puncture and serum was obtained by centrifugation at low speed. Samples were either frozen to obtain DNA or preserved in RNA later for RNA extraction. Four animals per experimental group were used.

### 4.3. DNA Extraction

DNA from peripheral blood lymphocytes was extracted using the Qiagen DNA Blood MiniKit (Qiagen, Hilden, Germany), doing a first erythrocytes hemolysis step and following manufacturer’s specifications. DNA from the brain and liver of mice was extracted using the Qiagen DNA Mini Kit (Qiagen), following manufacturer’s instructions. DNA from the serum of mice was obtained with FitAmp Plasma/Serum DNA Isolation Kit (Epigentek, New York, NY, USA) according to manufacturer’s instructions.

In all cases, quality and concentration of DNA was measured with an Epoch Microplate Spectrophotometer. Only DNA samples with 260/280 and 260/230 ratios above 1.8 were used for this study.

### 4.4. RNA Extraction

Total RNA was extracted from peripheral blood lymphocytes using the PureLink™ RNA Mini Kit (Thermo Fisher Scientific, Waltham, MA, USA) following specifications from the manufacturer. Total liver’s mice RNA was obtained using a RNAEasy mini Kit (Qiagen, Hilden, Germany), following manufacturer’s instructions. Briefly, samples were centrifuged to eliminate Qiazol reagent and lysed in the presence of lysis buffer and 2-β mercaptoethanol. Lysates were then transferred to purification columns and treated with Pure-Link™ DNAse (Thermo Fisher Scientific, Waltham, MA, USA). After several washing steps, RNA was eluted with RNAse free water. RNA concentration and purity were measured with an Epoch Microplate Spectrophotometer (BioTek instruments, Bad Friedrichshall, Germany). Only RNA samples with 260/280 and 260/230 ratios above 1.8 were used for this study.

### 4.5. Quantification of Global DNA Methylation (5mC) and Hydroxymethylation (5hmC)

The global 5mC study included 40 patients with AD, 35 with PD, 26 with DV and 40 healthy controls. For the 5hmC study, we used 47 patients with AD, 30 with PD, 21 with DV and 40 healthy controls. Characteristics of these patients are shown in Table 1. Global 5mC and 5hmC levels were measured by an ELISA-like colorimetric methylated or hydroximethylated DNA quantification kit (Epigentek, New York, NY, USA), following manufacturer’s instructions. Briefly, 100 ng DNA was used, and after different reactions and washes, absorbance was measured at 450 nm with an Epoch Microplate Reader (BioTek instruments, Bad Friedrichshall, Germany). Results are expressed as mean ± S.E.M. of the percentage of 5mC or 5hmC, according to the manufacturer’s manual. Briefly, to quantify the absolute amount of methylated DNA, we generated a standard curve using linear regression function with Microsoft Excel. The amount and percentage of 5mC was calculated with the formula:5mC (ng) = (Sample OD − Blank OD)/(Slope × 2)(1)
5mC (%) = 5mC (ng)/sample DNA (ng) × 100(2)

Global 5mC study in brain and liver’s samples from wild type and transgenic mice was analyzed with the same protocol as the one used with blood samples.

### 4.6. Quantitative Real Time RT-PCR

This study includes seven healthy subjects, 10 patients with PD and eight with different types of dementia. Characteristics of these patients are shown in Table 1. RNA was reverse-transcribed following specifications of the High Capacity cDNA Reverse Transcription Kit (Thermo Fisher Scientific, Waltham, MA, USA). In total, 400 ng of RNA was used for the retrotranscription reaction and the following thermocycling conditions were used: 10 min at 25 °C, then 120 min at 37 °C and 5 min at 85 °C.

DNMT1 and DNMT3a expression was quantified by qPCR using the StepOne Plus Real Time PCR system (Thermo Fisher Scientific, Waltham, MA, USA), following manufacturer’s instructions. PCR was carried out in duplicates using the Taqman Gene Expression Master Mix (NZYTech, Lisboa, Portugal) and specific TaqMan probes (Thermo Fisher Scientific, Waltham, MA, USA) for human DNMT1 (Assay ID Hs1027162_m1) and DNMT3a (Assay ID Hs00945875_m1) and mice DNMT3a (Assay ID Hm00432881_m1). Relative quantification was done using the comparative CT method [76] with the StepOne Plus Real Time PCR software and are expressed as fold induction with respect to healthy samples. Data were corrected with Human Glyceraldehyde-3-Phosphate Dehydrogenase GAPDH (Assay ID Hs02786624_g1) or mice S18 (Assay ID Mm03928990_g1) mRNA levels. Results are shown as mean ± S.E.M.

### 4.7. Correlation Analysis

Analysis of correlation between 5mC and 5hmC levels was performed with regression studies in Microsoft Excell software. R2 value is indicated in graphics. *p*-values are also indicated in graphics when this value is significant or close to significant.

### 4.8. Statistical Analysis

Shapiro-Wilk test was used to check a normal distribution and Levene test to determine the equality of variances. As the samples followed a normal distribution, statistical significance was determined using the SPSS program by applying ANOVA test and Bonferroni test. Results are expressed as means ± S.E.M. The *p*-value is indicated in the figures as * *p* < 0.05, ** *p* < 0.01 and *** *p* < 0.001.

## 5. Conclusions

This is a study about the methylation changes in patients with Alzheimer’s disease, Parkinson’s disease and Cerebrovascular disorders. We observed a significant decrease in 5mC and 5hmC levels in the three pathologies. DNMT3a expression was also found to be downregulated in patients with different types of Dementia. Finally, in an Alzheimer’s transgenic mice model, we observed a decrease in 5mC levels in brain, liver and serum samples. DNMT3a expression was also found to be reduced in the brain of transgenic AD mice. All these results suggest that the combination of the measurement of global DNA methylation and DNMT3a expression, used in combination with other routine clinical proofs, might represent a novel biomarker for age-related brain disorders in specialized settings.

## Figures and Tables

**Figure 1 ijms-21-02220-f001:**
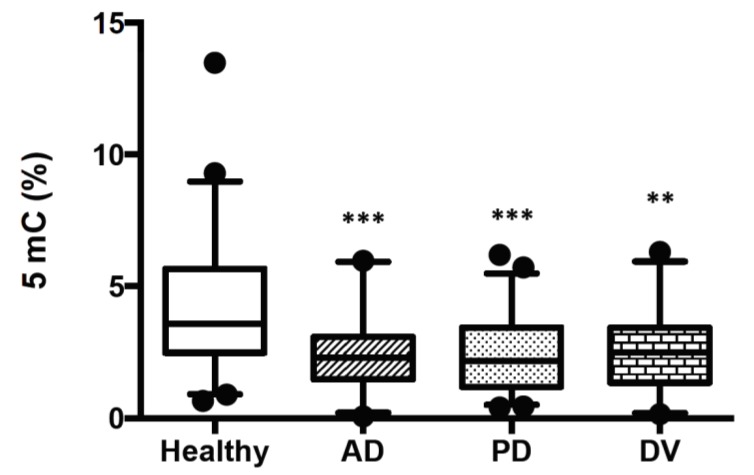
5mC levels in blood DNA samples from healthy subjects and patients with AD, PD or DV. DNA global methylation was determined with an Enzyme-Linked ImmunoSorbent Assay (ELISA)-like colorimetric assay as described in Material and Methods. The study includes 40 patients with AD, 35 with PD, 26 with DV and 40 healthy controls. Data are shown in a box-plot graphic. Statistical significance between groups was calculated with ANOVA and the Bonferroni test and is shown as ** *p* < 0.01 and *** *p* < 0.001.

**Figure 2 ijms-21-02220-f002:**
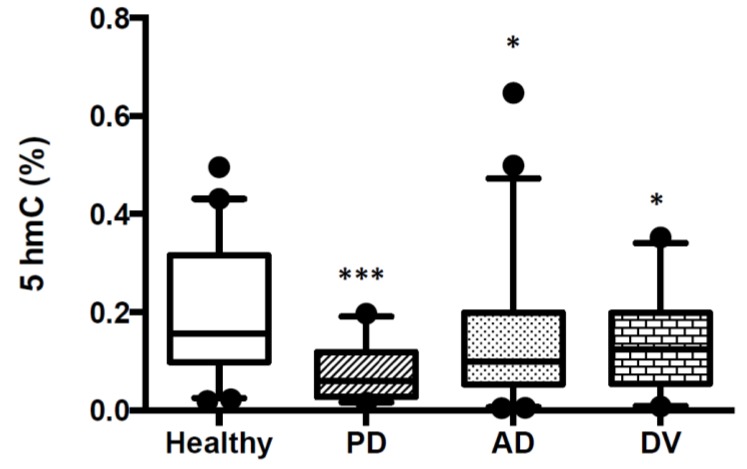
5hmC levels in blood DNA samples from healthy subjects and patients with AD, PD or DV. Levels of DNA global hydroxymethylation were assayed with an ELISA-like colorimetric assay as described in the Material and Methods. The study includes 47 patients with AD, 30 with PD, 21 with DV and 40 healthy controls. Results are shown in a box-plot graphic. Significance of analysis of variance (ANOVA) and Bonferroni post-test among the indicated groups is shown as * *p* < 0.05 and *** *p* < 0.001.

**Figure 3 ijms-21-02220-f003:**
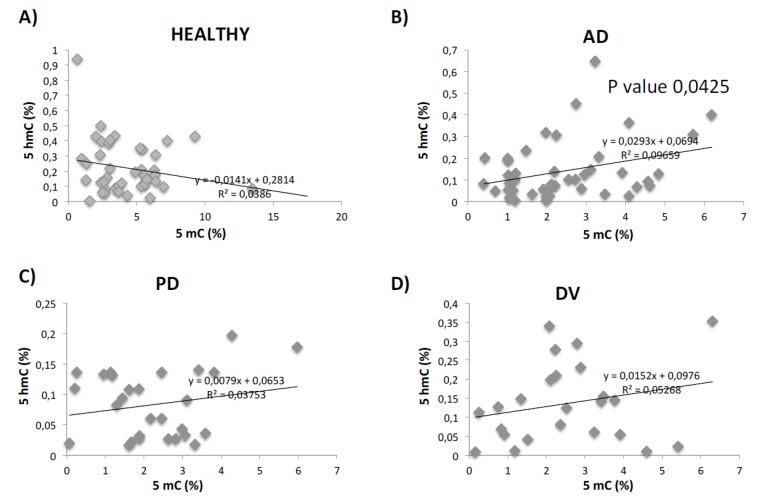
Correlation analysis between 5mC and 5hmC levels. Regression analysis of 5mC and 5hmC values in healthy (**A**), AD (**B**), PD (**C**) and DV (**D**) groups. *p*-values are also indicated in the graphics when this value is significant. *p*-value is indicated in the figures as * *p* < 0.05.

**Figure 4 ijms-21-02220-f004:**
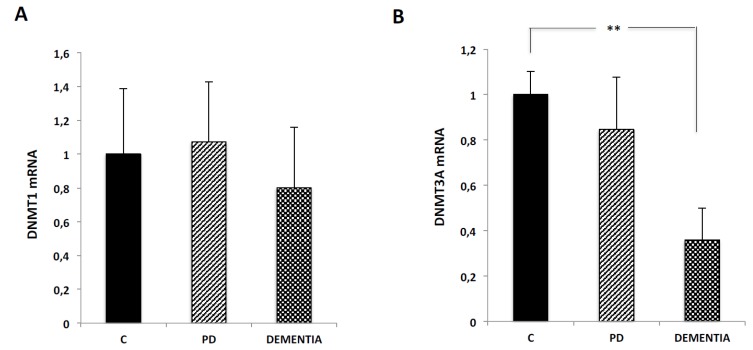
DNMT expression in blood samples from healthy subjects and patients with PD or dementia. Real time RT-qPCR in samples of the different groups with Taqman probes for DNMT1 (**A**) and DNMT3a (**B**). The study includes seven healthy subjects, 10 patients with PD and eight patients with different types of dementia. All data are means ± S.E.M. and are expressed relative to the values obtained in samples from healthy subjects. Significance of ANOVA and Bonferroni post-test among the indicated groups is shown as ** *p* < 0.01.

**Figure 5 ijms-21-02220-f005:**
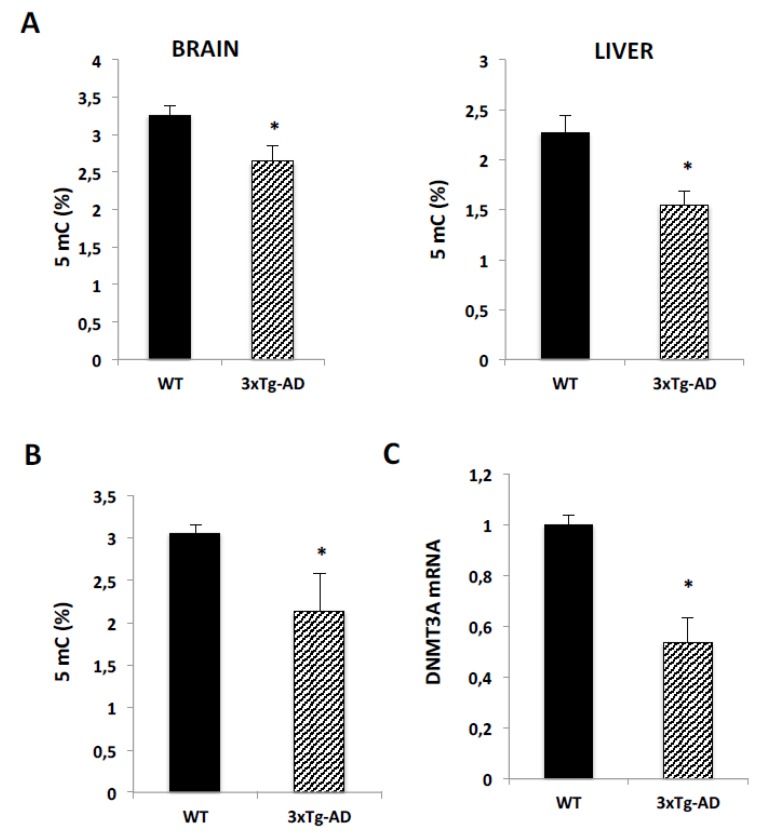
Methylation status study in a 3xTg-AD mice model. (**A**) Global DNA methylation was quantified with an ELISA-like colorimetric kit in brain (left) and liver (right) samples in wild type and 3xTg-AD mice. (**B**) Similar to (A) in serum mice samples. (**C**) Real time RT-qPCR in brain samples of wild type and triple transgenic mice was performed with Taqman probes DNMT3a. All data are means ± S.E.M. and are expressed relative to the values obtained in wild type mice. Significance of ANOVA and Bonferroni post-test among the indicated groups is shown as * *p* < 0.05.

**Table 1 ijms-21-02220-t001:** Patient’s characteristics.

	Population Characteristics						
5 hmC study (Figure 1 and Figure 3)			**Total**	**Healthy**	**AD**	**PD**	**DV**
	Gender	N	141	40	40	35	26
		Age		66.61 ± 10.11	68.18 ± 9.48	69.24 ± 6.23	72.15 ± 6.46
		Male		26	19	20	11
		Female		17	26	17	15
	APOE Genotype	3.3		30	14	27	12
		3.4		9	17	4	12
		4.4			8		
		2.3		1	1	3	1
		2,4				1	1
5 hmC study (Figure 2 and Figure 3)
	Gender	N	131	40	47	30	21
		Age		63.14 ± 10.11	66.63 ± 12.76	70.24 ± 6.75	71.77 ± 6.8
		Male		25	20	18	9
		Female		15	27	12	11
	APOE Genotype	3.3		29	16	24	9
		3.4		9	20	3	10
		4.4			9		
		2.3		2	1	2	1
		2,4				1	1
DNMTs expression study (Figure 4)
	Gender	N	25	7	10	8	
		Age		67.71 ± 8.28	67.9 ± 6.1	79.89 ± 5.06	
		Male		3	6	3	
		Female		4	4	5	
	APOE Genotype	3.3		4	6	2	
		3.4		1	1	5	
		4.4					
		2.3		1	3	1	
		2,4				1	
DNMTs expression study (Figure 4)	PATIENT	DIAGNOSIS					
	1	Mixed Dementia/AD					
	2	Mixed Dementia/Ischemic Vascular Encephalophaty					
	3	Senile Dementia/Mixed Dementia/Multi-infarction Vascular Encephalophaty					
	4	Mixed Dementia/Bingswanger Disease					
	5	Mixed Dementia Aphasic variety					
	6	Senile Dementia/Toxic Dementia					
	7	Ischemic Encephalophaty					
	8	Senile Dementia					

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
