# Peer review of "DNA Methylation in Neurodegenerative and Cerebrovascular Disorders"

_ijms, 2020, doi:10.3390/ijms21062220_

Round 1

Reviewer 1 Report

This is a comparative study of global DNA methylation, hydroxymethylation and DNMT levels in normal control versus AD, PD and DV samples. The authors show reduced global methylation and hydroxymethylation in the three pathological groups compared to the normal controls. The sample sizes in used in the three groups is a concern and the results need to be validated in some larger cohorts.

Following are some suggestions for the revised version.

It would be good to mention the tissue used and the sample sizes in the abstract. The tissue is mentioned only in the title of the results section. It would be better to mention this under the figures as well.

The authors mention that clinical diagnosis of the patients was done based on thorough clinical and genomic examination. It is important to describe the methodology in detail or references to previous publications. As mentioned in table 1, the authors could discuss the heterogeneity of pathological diagnosis of samples in relation to the results and conclusion.

It is not clear how the quantification of 5 mC, 5 hmC and DNMT levels were done. Are there any normalisation methods used for this? The authors could cite some references for this. From the figures 1 and 2, it seems quantification was done as percentages but it is not clear how these percentages were derived. In this regard the statistical methods used for the analyses using percentages may not be appropriate. It would be good to present the histogram of the data in the three groups in supplementary material. The authors mention about the normality test in the methods section but are there no results about normality tests.

The authors mention about assessment of sample characteristics such as age, sex etc., with the methylation levels but have not shown the results. I would suggest to present the results in supplementary material with the description of the methods.

It is important to use minimal covariates such as age and sex for comparison of the groups. The authors could perform either analysis of covariance or regression (if necessary, with appropriate transformation for the 5 mC, 5 hmC and DNMT levels) with the covariates age, sex and apoe 4 carrier status and present the complete results including the actual p-values.

It is not clear if the same set of samples were used for 5 mC, 5 hmC and DNMT studies. If these were measured on the same samples some correlation analysis could be done among these three measures. Also, if would be interesting to examine the combined influence of these two measures in relation to the four groups.

Author Response

Thank you very much for your careful review of the manuscript. We have tried to incorporate your suggestions and comments in a new version of the manuscript.

1.- This is a comparative study of global DNA methylation, hydroxymethylation and DNMT levels in normal control versus AD, PD and DV samples. The authors show reduced global methylation and hydroxymethylation in the three pathological groups compared to the normal controls. The sample sizes in used in the three groups is a concern and the results need to be validated in some larger cohorts.

This study includes 141 samples for 5 mC analysis, 131 for 5 hmC study and 25 samples for DNMTs expression studies. Our results suggest that DNA methylation plays an important role in neurodegeneration. In our opinion the number of samples analyzed is enough for a first study. However we agree with the reviewer and we will extend the study in a lager sample size in a future project.

2.- It would be good to mention the tissue used and the sample sizes in the abstract. The tissue is mentioned only in the title of the results section. It would be better to mention this under the figures as well.

Following reviewer advice we have added information about tissues used and sample size in the abstract (lines 14-15, 18-20). We have also incorporated this information in figure legends (line 88-89, 119-120, 160-161).

3.- The authors mention that clinical diagnosis of the patients was done based on thorough clinical and genomic examination. It is important to describe the methodology in detail or references to previous publications. As mentioned in table 1, the authors could discuss the heterogeneity of pathological diagnosis of samples in relation to the results and conclusion.

Following the reviewer request we have clarified this point in material and methods section (lines 336-337). As reviewer suggest we have discussed the heterogeneity of pathological diagnosis and its importance in our results (line 208-211, 223-224).

4.- It is not clear how the quantification of 5 mC, 5 hmC and DNMT levels were done. Are there any normalisation methods used for this? The authors could cite some references for this. From the figures 1 and 2, it seems quantification was done as percentages but it is not clear how these percentages were derived. In this regard the statistical methods used for the analyses using percentages may not be appropriate. It would be good to present the histogram of the data in the three groups in supplementary material. The authors mention about the normality test in the methods section but are there no results about normality tests.

As the reviewer suggests we have detailed the quantification of 5 mC, 5 hmC and DNMT levels in material and methods section (lines 382-388, 402-405). We have added a reference about 5 mC quantification (Galpern W et al, Ann Neurol, 2006) and a reference about mRNA expression analysis (Wong M.L et al, BioTecnbiques, 2005). We have incorporated the histograms of the data in figures supl 1 and 2 and in the results section (line 83-84, 113-114). Finally, we have clarified the statistical analysis section of material and methods sections (line 413-414) adding a mention about normality test results.

5 and 6- The authors mention about assessment of sample characteristics such as age, sex etc., with the methylation levels but have not shown the results. I would suggest to present the results in supplementary material with the description of the methods.

It is important to use minimal covariates such as age and sex for comparison of the groups. The authors could perform either analysis of covariance or regression (if necessary, with appropriate transformation for the 5 mC, 5 hmC and DNMT levels) with the covariates age, sex and apoe 4 carrier status and present the complete results including the actual p-values.

Following the reviewer suggestion we have incorporated the correlation analysis of 5 mC and 5 hmC values with age, sex… in fig supl 3-10. We have incorporated a section about supplementary information with the description of material and methods and figure legends. We have described these results in results  (line 96-100, 102-103, 124-127, 149-155) and discussion (lines 234-237, 239-242, 246-250, 278-283) section.

7.-It is not clear if the same set of samples were used for 5 mC, 5 hmC and DNMT studies. If these were measured on the same samples some correlation analysis could be done among these three measures. Also, if would be interesting to examine the combined influence of these two measures in relation to the four groups.

Answering to the reviewer question, we have used similar samples for 5 mC and 5 hmC studies. The number of samples of each group (healthy, AD, PD and DV) is different in 5 mC than in 5 hmC due to problems with the amount of sample or other methodological issues.  We have analyzed the correlation between 5 mC and 5 hmC levels and we only observed a significant correlation in AD. We have incorporated these results in Figure 3. Thus, we have changed the numbers of Fig 3 and 4 to 4 and 5 respectively. This information is included in results (line 128-133) section. Finally, we have added the description of correlation analysis in material and methods section (lines 407-410) and incorporated a figure legend description for this figure at the end of the manuscript. We have also improved the information in table 1.

            On the other hand, in DNMT studies we used different samples. The CIBE collection has incorporated later samples conserved in Qiazol Lysis Reagent (Qiagen) so that we have a smaller number of samples for DNMTs expression analysis than for 5 mC or 5 hmC studies. We have clarified this point in material and methods section (lines 342-343).

Reviewer 2 Report

I found multiple reviews of studies of DNA methylation (global and on the level of CpG sites) in neurodegenerative and age-related diseases. Global DNA methylation in peripheral blood cannot serve as a biomarker of specific neurodegenerative diseases. There are multiple diseases (for example some immune diseases) that show the same tendencies in global 5mC and 5 hmC levels as described in the manuscript. 

It is not clear if blood from the same participants' buffy coat had been used for 5 mC, 5 hmC levels and DNMT expression measurements. It looks like different participants donated blood for these assays. It would make more sense to use blood samples from the same participants for all assays, and this design would allow correlation of results.

Author Response

Thank you very much for your revision and comments. We have tried to incorporate your points in a new version of the manuscript.

I found multiple reviews of studies of DNA methylation (global and on the level of CpG sites) in neurodegenerative and age-related diseases. Global DNA methylation in peripheral blood cannot serve as a biomarker of specific neurodegenerative diseases. There are multiple diseases (for example some immune diseases) that show the same tendencies in global 5mC and 5 hmC levels as described in the manuscript.

We agree with the reviewer in that DNA methylation is also regulated in other pathologies. We have clarified this point in discussion (line 322-327) and conclusions (424-425) sections. Our results indicate that DNA methylation is regulated in neurodegenerative and cerebrovascular diseases and we propose this proof as a complementary test to the patient, in combination with another clinical proofs such as brain mapping, neuroimaging proofs…

It is not clear if blood from the same participants' buffy coat had been used for 5 mC, 5 hmC levels and DNMT expression measurements. It looks like different participants donated blood for these assays. It would make more sense to use blood samples from the same participants for all assays, and this design would allow correlation of results.

Answering to the reviewer question, we have used similar samples for 5 mC and 5 hmC studies. The number of samples of each group (healthy, AD, PD and DV) is different in 5 mC than in 5 hmC due to problems with the amount of sample or other methodological issues.  We have analyzed the correlation between 5 mC and 5 hmC levels and we only observed a significant correlation in AD. We have incorporated these results in Figure 3. Thus, we have changed the numbers of Fig 3 and 4 to 4 and 5 respectively. This information is included in results (line 128-133) section. Finally, we have added the description of correlation analysis in material and methods section (lines 407-410) and incorporated a figure legend description for this figure at the end of the manuscript. We have also improved the information in table 1.

            On the other hand, in DNMT studies we used different samples. The CIBE collection has incorporated later samples conserved in Qiazol Lysis Reagent (Qiagen) so that we have a smaller number of samples for DNMTs expression analysis than for 5 mC or 5 hmC studies. We have clarified this point in material and methods section (lines 342-343).

Round 2

Reviewer 1 Report

Minor suggestions

  1. APOE genotype frequencies under 5 hmC study can be removed as it is left empty and is the same as 5 mC study.
  2. To be consistent, all the decimal places can be indicated by "." instead of ",".
  3. Include normality test results in the supplementary material.

Author Response

Thank you very much for your comments. We have tried to incorporate your suggestions and comments in a new version of the manuscript.

1.- APOE genotype frequencies under 5 hmC study can be removed as it is left empty and is the same as 5 mC study.

We apologize by our mistake in Table 1. APOE genotype frequencies are slighty different in 5 mC and 5 hmC studies. We have changed the table by another with all the correct information of both studies.

2.- To be consistent, all the decimal places can be indicated by "." instead of ",".

Following reviewer advice the Text and Table 1 have been reviewed and decimal places have been homogenized as “.”.

3.- Include normality test results in the supplementary material.

Just as the reviewer requests we have included a new suplementary figure (Fig Supl3) with normality test results. We have also included this information in results section in the text (highlighted in green) and we have included its figure legend in supplementary information. Because of this change fig supl 3-11 changes to fig supl 4-12.

Reviewer 2 Report

The authors improved this manuscript significantly.

I still don't agree that DNA methylation can be used as a biomarker (with or without other measurements). However, I think that should be judged by readers.

Author Response

ANSWER TO REVIEWER 2

Thank you very much for your comments about our manuscript and for giving the opportunity that readers judge for themselves if DNA methylation could be a new biomarker.